# DualFed: Enjoying both Generalization and Personalization in Federated Learning via Hierachical Representations

## ABSTRACT

In personalized federated learning (PFL), it is widely recognized that achieving both high model generalization and effective personalization poses a significant challenge due to their conflicting nature. As a result, existing PFL methods can only manage a trade-off between these two objectives. This raises an interesting question: Is it feasible to develop a model capable of achieving both objectives simultaneously? Our paper presents an affirmative answer, and the key lies in the observation that deep models inherently exhibit hierarchical architectures, which produce representations with various levels of generalization and personalization at different stages. A straightforward approach stemming from this observation is to select multiple representations from these layers and combine them to concurrently achieve generalization and personalization. However, the number of candidate representations is commonly huge, which makes this method infeasible due to high computational costs. To address this problem, we propose DualFed, a new method that can directly yield dual representations correspond to generalization and personalization respectively, thereby simplifying the optimization task. Specifically, DualFed inserts a personalized projection network between the encoder and classifier. The pre-projection representations are able to capture generalized information shareable across clients, and the post-projection representations are effective to capture task-specific information on local clients. This design minimizes the mutual interference between generalization and personalization, thereby achieving a win-win situation. Extensive experiments show that DualFed can outperform other FL methods.

## CCS CONCEPTS

• **Computing methodologies → Distributed artificial intelligence**.

## KEYWORDS

Federated Learning, Generalization, Personalization, Representation Learning

## 1 INTRODUCTION

Federated learning (FL) [41] is an emerging machine learning paradigm that enables multiple clients to collaboratively train a model while preserving their data privacy. In real-world applications, data distributions across clients are often non-independent and identically distributed (Non-IID). For instance, in video surveillance, the data collected by distributed cameras can vary significantly due to differences in weather and lighting conditions [7, 19, 28, 42]. This Non-IID data distributions can significantly degrade the FL model performance [64, 67]. Currently, there are primarily two objectives to mitigate this issue: improving model generalization to accommodate more clients or enhancing model personalization to better adapt local data distributions. However, since local data

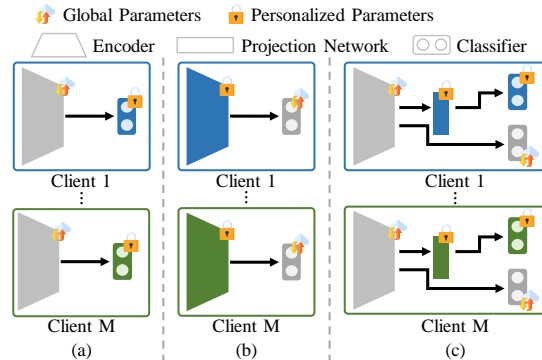

**Figure 1: Different forms that combines the representations and the classifier. (a) Global encoder with personalized classifier, (b) Personalized classifier with global encoder, (c) Our proposed DualFed that utilizes hierachical representations.**

distributions often differ from the global distribution in Non-IID FL, these two optimized objectives are typically in conflict.

Personalized federated learning (PFL), which aims to balance model generalization with personalization, serves as an effective approach to address the challenges posed by Non-IID data. Earlier PFL approaches suggest sharing the classifier or encoder, while personalizing the other [1, 12, 35]. This strategy aims to strike a balance between client collaboration and local adaptation, as presented in Figure 1 (a) and (b). However, these approaches can only ensure that the encoder to generate either the generalized or personalized representations. Thereby, some PFL methods suggest personalizing specific parameters within the encoder, allowing it to extract the representations that exhibit both generalization and personalization [33, 50, 52]. Additionally, some PFL techniques concurrently use global and personalized classifiers for predictions [6, 63] to harmonize generalization and personalization. Nevertheless, these methods inherently involve a trade-off between model generalization and personalization. This leads to an interesting question: *Is it feasible to create a model that can achieve both of these objectives concurrently in Non-IID FL?*

In fact, the dilemma in existed PFL methods primarily because they rely solely on post-encoder representations for decision-making. This design presents a significant hurdle as it necessitates the post-encoder representations to simultaneously exhibit both high generalization and personalization – objectives that are inherently contradictory in Non-IID FL. It is well known that deep models naturally produce hierarchical representations, as evidenced in studies such as [2, 18, 40, 44, 47, 51, 56, 59, 61]. The shallow layers capture general patterns that are transferable across different data distributions. As we delve into deeper layers, the representations become more specified for the downstream task. This implies that both the generalization and personalization that PFL seeks for are already

existed within the model. These observations do shed some lights on us: ***Can we leverage the hierachical representations within the deep model to achieve both high model generalization and personalization simulatenously?***

In this paper, we provide a positive response to the question posed earlier. A straightforward method for leveraging hierarchical representations involves directly selecting both generalized and personalized representations from them. However, this approach can incur substantial computational costs, owing to the volume of the candidate representations [48]. To address this problem, we introduce DualFed, a new PFL approach that not only straightforward to implement but also effectively decouple these two types of representations. As shown in Figure 1 (c), in DualFed, we modify the commonly used encoder-classifier architecture by inserting a projection network between the encoder and classifier. This modification generates representations at two distinct stages, aligning with the objectives of generalization and personalization, respectively. Specifically, the ***pre-projection representations*** generated before the projection network, are isolated from local tasks, making them more transferable across clients. Conversely, the ***post-projection representations*** produced after the projection network are closer to the decision layers, being more discriminative and personalized to local data distributions. To align with the objectives of these two representations, we maintain a shared encoder while localizing the projection network. A global classifier and a personalized classifier are trained using the pre-projection and post-projection representations, respectively. During inference, the outputs from these two classifier are combined to yield the final predictions, effectively benefiting from collaboration across clients and local adaptation.

We conduct extensive experiments on multiple datasets to demonstrate the effectiveness of DualFed. The experimental results show that DualFed can outperform state-of-the-art (SOTA) FL methods.

## 2 RELATED WORK

**Federated Learning.** FL [22, 30] can be categorized into general FL (GFL) [23, 31, 41] and personalized FL (PFL) [1, 12, 33, 35, 52, 53]. GFL aims to develop a generalized model that can be shared across clients. However, in Non-IID FL, it becomes challenging for a global model to satisfy the diverse needs of multiple clients, often leading to significant performance degradation [64, 67]. Consequently, PFL has emerged as an effective solution for these Non-IID situations by introducing model personalization to better align with local data distributions. There are various approaches to implement PFL, including model clustering [3, 4, 16, 49], and the personalization of specific parameters within the model [1, 12, 33, 35, 52]. However, these PFL methods can only manage a trade-off between these two objectives, as they expect the post-encoder representations to achieve the conflicting objectives.

**Representation Learning in Deep Models.** Since advanced deep learning models are typically organized as hierachical layers, analysing how representations evolve during the representation extraction process has been an established field [40, 44, 56, 59, 61]. Previous research indicates that deep models start by extracting generalized features and progressively filter out irrelevant components, retaining only those crucial for downstream tasks [40, 59]. This

has inspired numerous studies that leverage intermediate representations, in domains like object detection [36], image classification [14], and speech processing [11, 27]. However, selecting the optimal representations for each specific problem is computationally challenging [48]. In response, SimCLR [8] proposes to use a scalable projection network during training and discard it afterwards. This design has become a common practice in both supervised learning [15, 24, 57] and self-supervised learning [5, 9, 10, 17, 60]. Since then, numerous studies have explored the projector's role in model training from empirical [2, 32, 48, 57] and theoretical perspectives [21, 56, 58]. The common explanation is that the projection network differentiates the representations of the pre-training and downstream tasks, thereby enhancing the model transferability [57]. This situation is especially significant when the pre-training and downstream tasks are misaligned [2]. Nevertheless, the effects of projection network within FL are still not fully understood.

**Federated Learning within Representation Space.** The primary contribution of these methods is the regularization of the representation space to mitigate data heterogeneity [37, 38, 46, 54, 62, 63, 65, 66]. A straightforward strategy in these approaches involves directly calibrating the representation space. For instance, CCVR [38] post-calibrates the classifier after federated training using virtual representations. Another research direction links performance degradation to the misalignment of representation spaces across clients [65, 66]. In response, various methods have been developed to explicitly align the representation space across clients. Notably, FedProto [54], AlignFed [66], and FedFA [65] use class-wise representation centers for representation alignment. Additionally, some methods achieve alignment by implementing a fixed classifier. For instance, FedBABU [43] employs a randomly initialized classifier, SphereFed [13] introduces an orthogonal classifier, while FedETF [34] implements an ETF (Equiangular Tight Frame) classifier during model training. However, these representation alignment methods primarily focus on extracting generalized representations shareable across clients, often overlooking the personalized representations specific to local tasks. Consequently, recent studies have concentrated on balancing both model generalization and personalization. For example, Fed-RoD [6] achieving this goal combining the predictions of personalized and global classifiers. Yet, these methods face challenges, as they rely solely on representations at the same stage. Expecting the single-stage representations to exhibit both generalization and personalization is often intertwined.

## 3 PRELIMINARIES

In this section, we present some preliminaries related to DualFed, including the PFL framework and its motivation.

### 3.1 Federated Learning

In this paper, we consider a standard PFL setting which consists of a central server and $M$ distributed clients. For each client $m \in [M]$, there are totally $N_m$ samples $\{\boldsymbol{x}_m^i, \boldsymbol{y}_m^i\}_{i=1}^{N_m}$ drawn from the distribution $\mathcal{D}_m$, where $\boldsymbol{x}_m^i \in \mathcal{X}_m \subseteq \mathbb{R}^n$ represents the raw input and $\boldsymbol{y}_m^i \in \mathcal{Y}_m \subseteq \{0, 1\}^C$ represents the corresponding label, with $C$ denoting the total number of classes. In Non-IID scenarios within PFL, the data distributions are assumed to be heterogeneous across clients, indicating that $\mathcal{D}_i \neq \mathcal{D}_j, \forall i, j \in \{1, 2, \dots M\}, i \neq j$.

The goal of a standard PFL setting is to develop a model $\psi_m(\cdot)$ parameterized by $\Theta_m$ for client $m$. The corresponding optimization objective can be expressed as:

$$\arg \min_{\Theta_1,\dots,\Theta_M} \mathcal{L}(\Theta_1,\dots,\Theta_M) \triangleq \arg \min_{\Theta_1,\dots,\Theta_M} \frac{1}{M} \sum_{m=1}^{M} \mathcal{L}_m(\Theta_m), \quad (1)$$

where $\mathcal{L}(\Theta_1,\dots,\Theta_M)$ represents the overall optimization objective for the PFL system, $\mathcal{L}_m(\Theta_m)$ denotes the empirical risk for client $m$. In PFL, directly optimizing $\mathcal{L}(\Theta_1,\dots,\Theta_M)$ is commonly infeasible as the clients cannot access the data on other clients. Therefore, a PFL training procedure typically involves the independently local updating performed on participating clients utilizing their own empirical risk and the model aggregation performed on the server. Specifically, for client $m$, its empirical risk is defined as:

$$\mathcal{L}_m(\Theta_m) := \frac{1}{N_m} \sum_{i=1}^{N_m} \ell(\boldsymbol{y}_m^i, \hat{\boldsymbol{y}}_m^i), \quad (2)$$

with $\hat{\boldsymbol{y}}_m^i = \psi_m(\boldsymbol{x}_m^i; \Theta_m)$ representing the model's prediction for $\boldsymbol{x}_m^i$, and $\ell : \mathcal{Y} \times \mathcal{Y} \to \mathbb{R}$ being the loss function that quantifies the prediction error (e.g., cross-entropy loss).

Once the local training on clients is completed, the participating clients upload their updated global parameters within the model to the server. The server then averages the parameters at corresponding positions to generate new global parameters. These global parameters are subsequently distributed to the clients for the next round of local updating. By iteratively performing local training and model aggregation, PFL facilitates collaborative model training without the need to share raw data from the clients.

For the sake of brevity, we occasionally omit the superscript denoting the sample index in subsequent sections of this paper. Additionally, we sometimes denote personalized parameters with the superscrip $p$ (e.g., $\Theta_m^p$), and global parameters with the superscript $s$ (e.g., $\Theta_m^s$), to clarify the expressions in the following sections.

### 3.2 Motivation of DualFed

As shown Figure 1 (a) and (b), in previous studies of PFL, the model $\Theta_m$ is commonly divided into an encoder $f_m(\cdot)$ and a classifier $h_m(\cdot)$ [1, 12, 43, 65, 66], parameterized by $\theta_m^f$ and $\theta_m^h$, respectively. The encoder $f_m(\cdot) : \mathcal{X}_m \to \mathcal{Z}_m$ generally consists of a series of stacked convolutional layers. It maps the raw input $\boldsymbol{x}_m$ from $\mathcal{X}_m \subseteq \mathbb{R}^n$ into a representation space $\mathcal{Z}_m \subseteq \mathbb{R}^k$, which is denoted as $\boldsymbol{z}_m = f(\boldsymbol{x}_m; \theta_m^f)$. Here, $\boldsymbol{z}_m \in \mathcal{Z}_m$ denotes the representation generated from $\boldsymbol{x}_m$ utilizing the encoder $f_m(\cdot)$. Practically, the dimension of this representation is significantly smaller than that of the raw input, which implies that $k \ll n$. The classifier, $h_m(\cdot) : \mathcal{Z}_m \to \mathcal{Y}_m$, generally includes a fully connected (FC) layer and a softmax layer. It generates the normalized predictions $\hat{\boldsymbol{y}}_m$ based on the representation $\boldsymbol{z}_m$, which is indicated as $\hat{\boldsymbol{y}}_m = h_m(\boldsymbol{z}_m; \theta_m^h)$.

Nonetheless, the widely used encoder-classifier architecture proves to be problematic in PFL. In this framework, each client's model has dual objectives: collaborating with other clients to improve its generalization and adapting to its own local data distribution for better personalization. Within the encoder-classifier architecture, only the representations after the encoder, referred to as the ***post-encoder representations***, are used for decision-making.

This approach can lead to a dilemma in PFL, as generalization and personalization are contradictory objectives, particularly in Non-IID scenarios. More specifically, to enhance model generalization, the post-encoder representations should capture shared information across varying data distributions among clients. On the other hand, enhancing model personalization requires these representations to capture specific information aligned with each client's local data distribution. When the data distribution varies significantly across clients, these two types of information can be vastly different. Consequently, in this encoder-classifier architecture, ensuring that the post-encoder representations simultaneously meet these two conflicting objectives is a challenging task.

To address the dilemma mentioned earlier, we shift our focus on the process of representation extraction within the deep models. Advanced deep models are typically organized in a hierachical architecture. As shown in previous studies, these models initially extract generalized representations that are transferable across various data distributions [2, 18, 40, 44, 47, 51, 56, 59, 61]. As the model progresses to deeper layers, it gradually discards irrelevant components and retains only information relevant to the specific task. In other words, both the generalized and personalized representations that PFL seeks for are already existed within the model. By leveraging these hidden generalized and personalized representations, we can achieve both high generalization and personalization in PFL. However, directly extracting these specific representations during the representation extraction phase is computationally challenging [48]. Therefore, DualFed adopts a simpler strategy by incorporating a personalized projection network, which effectively decouples the generalized and personalized representations. Further details on this approach are discussed in the subsequent sections.

## 4 METHOD

In this section, we present a detailed discussion of our proposed DualFed. First, we provide a framework overview of DualFed. Following that, we present the procedure of local training on the clients and model aggregation implemented on the server, respectively.

### 4.1 Framework Overview of DualFed

Figure 2 presents the framework of DualFed. It aligns with the standard training framework of PFL, which includes iterative local training on clients and global model aggregation on the server. The key innovation in DualFed, as compared to previous PFL methods, is the integration of a personalized projection network situated between the encoder and the classifier. We refer to this personalized projection network as $g_m^p(\cdot)$, with its parameters denoted by $\theta_m^{g,p}$. Functionally, this projection network, $g_m^p(\cdot) : \mathcal{Z}_m \to \mathcal{U}_m$ is usually a MLP (multi-layer perceptron). By inserting this projection network, the representations produced by the encoder are not directly inputted into the classifier for prediction. Instead, they first pass through the projection network, which remaps them to a personalized representation space $\mathcal{U}_m \subseteq \mathbb{R}^d$. Formally, we represent this process as $\boldsymbol{u}_m = g(\boldsymbol{z}_m; \theta_m^{g,p})$. For clarity, we term the representation before the projection network (i.e., $\boldsymbol{z}_m$) as the ***pre-projection representations***, and the representation after the projection network (i.e., $\boldsymbol{u}_m$) as the ***post-projection representations***.

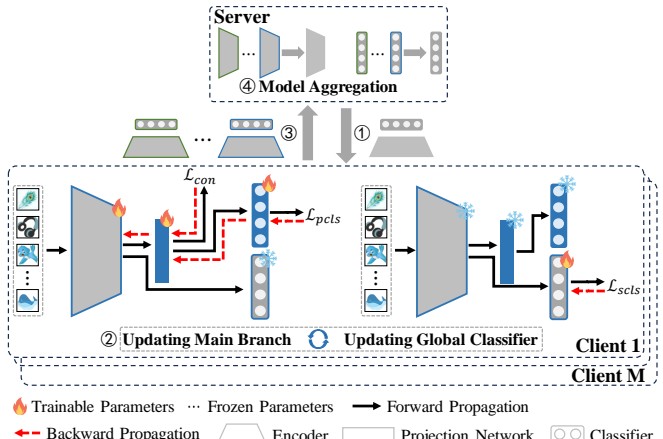

**Figure 2: Framework overview of DualFed. It consists of 4 steps in a single global round: 1) the server broadcasts global encoder and classifier to each client; 2) each client performs local updating by iteratively updaing main branch and global classifier; 3) each client uploads its updated global encoder and classifier to the server; 4) the server aggregates encoders and classifiers from clients to generate new ones. This process is repeated until the model converges.**

Drawing on the hierarchical nature of deep model representation extraction, the pre-projection and post-projection representations in our framework exhibit distinct characteristics, aligning with the generalized and personalized objectives of PFL, respectively. Specifically, the pre-projection representations are separated from the final outputs by the projector network, meaning that they are not directly tied to the local tasks on each client. As previously studies have shown, these pre-projection representations are easier transferred across different data distributions [48, 57]. Therefore, in DualFed, the post-projection representations are fed into a global classifier $h_m^s(\cdot)$, which is parameterized by $\theta_m^{h,s}$. Additionally, to encourage the encoder to extract more generalized information, we let the encoder be shared among clients in DualFed. The predictions from this global classifier is expressed as:

$$\hat{\boldsymbol{y}}_m^s = h_m^s \circ f_m^s(\boldsymbol{x}_m), \forall m \in [M]. \tag{3}$$

Conversely, the post-projection representations are more closely aligned with the final outputs. This implies that these representations are more pertinent to accomplishing tasks related to the local data distribution. In DualFed, to effectively adapt to these local distributions, we utilize a personalized classifier $h_m^p(\cdot)$ for each client, parameterized by $\theta_m^{h,p}$, to adapt to the local data distribution. For a given input $\boldsymbol{x}_m$, the prediction generated by this personalized classifier can be expressed as follows:

$$\hat{\boldsymbol{y}}_m^p = h_m^p \circ g_m^p \circ f_m^s(\boldsymbol{x}_m), \forall m \in [M]. \tag{4}$$

During inference, the final predictions are derived by ensembling the outputs from both the global classifier and the personalized classifier. This process is expressed as follows:

$$\hat{\boldsymbol{y}}_m = \hat{\boldsymbol{y}}_m^p + \hat{\boldsymbol{y}}_m^s, \forall m \in [M]. \tag{5}$$

By integrating a personalized projection network between the encoder and the classifier, DualFed effectively separates the contradictory optimization objectives inherent in PFL into distinct stages within the model. This approach resolves the conflict of pursuing contradictory objectives within the representations in the same stage, thereby can achieve a win-win situation between the model generalization and personalization.

### 4.2 Local Training on Client

In DualFed, each client updates the model for $E$ rounds using its own datasets after revceiving the global models from the sever. In order to fully exploit the hierarchical characteristics of deep model representations and achieve the optimization objectives of PFL, we introduce a stage-wise training procedure for local clients.

At the first stage, we freeze the global classifier and training the main branch of the model. The main branch comprises the global encoder, the personalized projector, and the personalized classifier, with their parameters collectively represented as $\overline{\Theta}_m :=$ $\{\theta_m^{f,s}, \theta_m^{g,p}, \theta_m^{h,p}\}$. This stage allows the model to extract both generalized and personalized representations. To ensure the model's effectiveness in accomplishing local tasks, we employ cross-entropy loss as the classification loss, as indicated in the following equation:

$$\mathcal{L}_{pcls} = \sum_{i=1}^{N_m} \sum_{c=1}^{C} \boldsymbol{y}_m^{i,c} \log(\hat{\boldsymbol{y}}_m^{p,i,c}), \tag{6}$$

where $\boldsymbol{y}_m^{i,c}$ denotes the value at $c^{th}$ class of the one-hot ground-truth label of the $i^{th}$ sample on client $m$, $\hat{\boldsymbol{y}}_m^{p,i,c}$ represents the normalized prediction probability of $c^{th}$ classes of the $i^{th}$ sample on client $m$ from the personalized classifier.

As the post-projection representations are tailored to adapt to the local data distribution, we further enhance its discrimination by implementing supervised contrastive loss [24], as demonstrated in the following equation:

$$\mathcal{L}_{con} = -\frac{1}{N_m} \sum_{i=1}^{N_m} \frac{1}{|A(i)|} \sum_{j \in A(i)} \log \frac{\exp(\boldsymbol{u}_i \odot \boldsymbol{u}_j/\tau)}{\sum_{a \in A \setminus \{i\}} \exp(\boldsymbol{u}_i \odot \boldsymbol{u}_a/\tau)} \tag{7}$$

where $A$ is the full set of samples, $A(i)$ consists of samples in $A$ that belong to the same class as $\boldsymbol{x}_m^i$, $\odot$ is the cosine similarity, and $\tau \in \mathcal{R}^+$ is the temperature coefficient.

The optimization objective at this stage is then defined as:

$$\overline{\Theta}_m^t = \arg\min_{\overline{\Theta}_m} \mathcal{L}_{pcls} + \lambda \mathcal{L}_{con}, \tag{8}$$

where $\lambda$ denotes the hyperparameter used for balancing these two loss terms, $t$ denotes the local updating epochs.

After updating $\overline{\Theta}_m$, we freeze its parameters and train the global classifier using the pre-projection representations to fulfill the local task, as represented in the following equation:

$$\mathcal{L}_{scls} = \sum_{i=1}^{N_m} \sum_{c=1}^{C} \boldsymbol{y}_m^{i,c} \log(\hat{\boldsymbol{y}}_m^{s,i,c}), \tag{9}$$

where $\boldsymbol{y}_m^{i,c}$ denotes the value at $c$ class of the one-hot ground-truth label of the $i_{th}$ sample on client $m$, $\hat{\boldsymbol{y}}_m^{s,i,c}$ represents the normalized prediction probability of $c$ classes of the $i_{th}$ sample on client $m$ from the global classifier.

The optimization objective in this stage can be expressed as:

$$\theta_m^{h,s,t} = \arg \min_{\theta_m^{h,s}} \mathcal{L}_{scls}. \tag{10}$$

In DualFed, both optimization objectives, as described in Eqs. (8) and (10), are optimized using mini-batch stochastic gradient descent (SGD). As evidenced by our experiments, this stage-wise optimization strategy diminishes the impact of local tasks on the pre-projection representations, thereby effectively preserving its generalization.

### 4.3 Model Aggregation on Server

Once the local updating process is complete, the clients send their global encoder and classifier parameters to the server. The server then aggregates these parameters using the following equation:

$$\tilde{\theta}^{f,s} = \sum_{m=1}^{M} \frac{1}{M} \theta_m^{f,s}, \quad \tilde{\theta}^{h,s} = \sum_{m=1}^{M} \frac{1}{M} \theta_m^{h,s}. \tag{11}$$

Following the model aggregation, the server broadcast the updated model back to the clients to initiate the subsequent round of local training.

## 5 EXPERIMENT

In this section, we conduct extensive experiments to showcase the effectiveness of DualFed, mainly including the comparative results with existed FL methods, and the additional analysis of DualFed.

### 5.1 Dataset Description

Our experiments are conducted on PACS [29], DomainNet [45], and OfficeHome [55]. The PACS dataset includes 4 distinct domains: Photo (P), Art Painting (A), Cartoon (C), and Sketch (S), each featuring images from 7 common categories. The DomainNet dataset encompasses 6 distinct domains: Clipart (C), Infograph (I), Painting (P), Quickdraw (Q), Real (R), and Sketch (S). Initially, each domain comprises 345 classes, but for our study, we narrow this down to 10 commonly used classes to create our experimental dataset. The Office-Home dataset contains images from 4 distinct domains: Art (A), Clipart (C), Product (P), and Real-World (R), each containing 65 classes. We retain all classes to conduct a comprehensive evaluation of DualFed on a larger-scale dataset.

For these datasets, we select the images from a single domain to form the dataset of an individual client. In both PACS and Domain-Net, we choose a subset of 500 training images per client from the same domain for the training dataset. For Office-Home, we set the number of training samples to 2,000 for the Clipart, Product, and Real-World domains. In the case of the Art domain, the number is limited to 1,942, matching the total number of samples available in this domain. All the images from the test dataset are reserved for evaluation for these datasets. We apply random flipping and rotational augmentations to these images during the training.

### 5.2 Compared Methods

We perform a comparative analysis against the following methods, including FedAvg[41], FedProx[31], FedPer[1], FedRep[12], LG-FedAvg[35], FedBN[33], FedProto[54], SphereFed[13], Fed-RoD[6], FedETF[34]. Additionally, the SingleSet method, where separate

models are trained and tested for each client using only their private data, is also used for comparison in our experiments.

### 5.3 Implementation Details

The adopted encoder is from the one of the ResNet18 model pretrained on the ImageNet dataset [20]. It is followed by a projector network, which consists of an FC network with the architecture: [Linear(512, 256) - ReLU - BN - Linear(256, 512) - BN]. To ensure uniform model capacity, all compared methods employ this Encoder-Projector architecture for representation extraction.

The learning rate is set 0.01, with a momentum of 0.5, for all methods except SphereFed . For SphereFed, we set the learning rate to 1.0 for Office-Home and to 0.1 for both DomainNet and PACS. During local updating, a batch size of 256 is consistent across all methods. The epoch of local updating is set to 1 for all methods except FedRep. For FedRep, it has a total of 5 local epochs, with the initial 4 epochs focusing on classifier optimization and the last epoch on encoder and projector optimization. The total global rounds is set to 300.

The other hyperparameters for different methods are selected by grid searching. To mitigate cross-domain interference and potential privacy issues related to BN layers, we localize the *running-mean* and *running-var* components within these layers for all methods.

To ensure the reliability of our results, each experiment is repeated 5 times with different random seeds: {0, 1, 2, 3, 4}. The subsequent sections will detail the mean and standard deviation of the highest test accuracy achieved during FL training. More implementation details are provided in Appendix.

### 5.4 Experimental Results

Tables 1 - 3 showcase the experimental results of our proposed DualFed alongside other FL methods on the PACS, DomainNet, and Office-Home datasets, respectively. Notably, DualFed presents a significant performance gain in comparison to these SOTA methods.

**Table 1: Experimental Results on PACS Dataset.**

| Method | P | A | C | S | Avg. |
|---|---|---|---|---|---|
| SingleSet | 97.78±0.56 | 88.12±0.25 | 89.19±0.37 | 91.01±0.73 | 91.52±0.10 |
| FedAvg | 97.72±0.56 | 89.24±1.01 | 89.32±0.60 | 91.01±0.70 | 91.82±0.34 |
| FedProx | 97.90±0.38 | 89.14±1.18 | 89.40±0.61 | 91.52±0.72 | 91.99±0.38 |
| FedPer | 98.20±0.42 | 89.54±1.16 | 91.28±0.75 | 91.29±0.60 | 92.58±0.57 |
| FedRep | 97.84±0.35 | 89.83±1.33 | 89.96±0.27 | 91.39±0.48 | 92.25±0.22 |
| LG-FedAvg | 97.60±0.54 | 88.46±0.45 | 89.74±0.30 | 91.36±0.66 | 91.79±0.24 |
| FedBN | 92.20±0.46 | 89.88±0.86 | 90.38±0.75 | 91.34±0.53 | 92.45±0.37 |
| FedProto | 97.90±0.19 | 91.15±0.50 | 92.22±0.61 | 92.99±0.59 | 93.57±0.34 |
| SphereFed | 98.26±0.35 | 88.95±0.87 | 91.11±0.42 | 91.03±0.82 | 92.34±0.26 |
| Fed-RoD | 98.02±0.36 | 88.85±1.04 | 89.79±0.49 | 90.85±0.59 | 91.88±0.31 |
| FedETF | 97.43±0.24 | 90.95±0.77 | 90.26±0.29 | 90.70±0.68 | 92.33±0.30 |
| DualFed | **98.32±0.24** | **92.47±0.42** | **94.91±0.63** | **94.32±0.61** | **95.01±0.31** |

Interestingly, the SingleSet model stands out as a strong benchmark, despite not collaborating with other clients. This is particularly evident in simpler domains, such as the Quickdraw domain in the DomainNet dataset. The underlying reason is that these simpler domains requires less complex semantic information for downstream tasks. In these cases, a personalized encoder's representations are sufficient, and collaboration for extensive semantic

**Table 2: Experimental Results on DomainNet Dataset.**

| Method | C | I | P | Q | R | S | Avg. |
|---|---|---|---|---|---|---|---|
| SingleSet | 88.25±0.81 | 50.99±1.24 | 89.60±1.00 | 82.78±0.43 | 94.07±0.12 | 88.16±0.53 | 82.31±0.19 |
| FedAvg | 89.47±0.97 | 53.70±0.86 | 89.60±0.52 | 80.58±0.80 | 92.85±0.52 | 88.56±0.58 | 82.46±0.33 |
| FedProx | 89.47±0.86 | 53.79±0.96 | 89.56±0.55 | 80.56±0.87 | 92.87±0.54 | 88.63±0.56 | 82.48±0.38 |
| FedPer | 89.70±0.81 | 54.22±0.68 | 92.12±0.98 | 82.18±0.65 | 94.76±0.41 | 89.57±0.66 | 83.76±0.32 |
| FedRep | 89.62±0.76 | 54.19±0.71 | 90.60±0.37 | 80.84±0.91 | 93.03±0.49 | 89.03±0.77 | 82.88±0.24 |
| LG-FedAvg | 88.56±0.83 | 51.54±1.18 | 89.89±0.78 | 82.68±0.74 | 94.20±0.32 | 88.59±0.70 | 82.58±0.09 |
| FedBN | 89.85±0.67 | 54.58±1.04 | 91.34±0.90 | 80.62±0.68 | 93.76±0.44 | 89.06±0.41 | 83.20±0.36 |
| FedProto | 90.04±0.86 | 54.31±0.91 | 92.18±0.55 | 84.82±0.67 | **94.82±0.25** | 90.40±0.56 | 84.43±0.30 |
| SphereFed | 88.97±0.52 | 51.02±1.63 | 90.69±0.43 | 78.50±1.18 | 92.65±0.33 | 88.77±0.54 | 81.77±0.48 |
| Fed-RoD | 89.70±0.99 | 52.91±0.89 | 90.18±0.51 | 81.64±0.50 | 93.03±0.46 | 88.88±0.73 | 82.72±0.25 |
| FedETF | 88.97±0.81 | 55.65±0.85 | 91.76±0.52 | 79.76±0.48 | 94.15±0.23 | 89.03±0.39 | 83.22±0.31 |
| DualFed | **92.51±0.41** | **56.77±0.95** | **94.41±0.30** | **85.18±0.30** | 94.69±0.08 | **92.27±0.54** | **86.14±0.12** |

extraction might be unnecessary or even detrimental. This observation is supported by LG-FedAvg's performance, which, while also utilizing a personalized encoder for representation extraction, outperforms SingleSet by leveraging collaborative training for a global classifier.

However, as the complexity within a domain increases, such as in the Infograph domain of DomainNet, the benefits of sharing the encoder among clients become apparent. This collaborative approach allows the encoder to extract more nuanced semantic information from the raw data, improving overall model performance, as demonstrated by the results of FedAvg and FedProx. FedRep and FedPer, employing a personalized classifier to adapt the representations from the global encoder, often outperform FedAvg and FedProx. However, these methods primarily leverage the global encoder's representations and do not fully utilize personalized information to cater to the local data distribution on individual clients.

FedProto significantly improves model performance by aligning representations from different clients within a unified representation space. Nonetheless, this alignment can result in a loss of semantic information pertinent to local tasks due to varying data distributions across clients. This issue is even more pronounced in models like SphereFed and FedETF, which employ a predefined classifier for representation alignment and lack specific semantic information about local data.

**Table 3: Experimental Results on Office-Home Dataset.**

| Method | A | C | P | R | Avg. |
|---|---|---|---|---|---|
| SingleSet | 66.52±1.27 | 74.27±0.60 | 87.46±1.02 | 77.54±0.58 | 76.45±0.32 |
| FedAvg | 68.82±1.30 | 74.91±1.02 | 85.82±0.36 | 80.30±0.53 | 77.46±0.35 |
| FedProx | 68.78±1.37 | 74.73±0.79 | 85.73±0.35 | 80.25±0.70 | 77.37±0.33 |
| FedPer | 70.31±1.07 | 75.03±0.38 | 87.76±0.18 | 80.51±0.43 | 78.40±0.40 |
| FedRep | 70.23±0.96 | 75.44±0.69 | 85.82±0.45 | 80.39±0.92 | 77.97±0.37 |
| LG-FedAvg | 67.22±1.30 | 75.33±0.19 | 87.44±0.43 | 77.80±0.27 | 76.94±0.26 |
| FedBN | 68.58±1.23 | 76.01±0.45 | 86.31±0.96 | 79.40±0.40 | 77.58±0.29 |
| FedProto | 67.92±0.74 | 75.76±0.57 | 87.80±0.30 | 77.89±0.41 | 77.34±0.25 |
| SphereFed | 66.68±0.89 | 69.12±0.82 | 81.92±0.95 | 76.76±0.28 | 73.62±0.48 |
| Fed-RoD | 68.21±0.86 | 75.42±0.37 | 86.40±0.72 | 80.30±0.79 | 77.58±0.22 |
| FedETF | 69.90±1.14 | 74.64±0.41 | 85.52±0.35 | 80.18±0.39 | 77.56±0.29 |
| DualFed | **71.01±0.71** | **77.41±0.47** | **88.84±0.47** | **81.70±0.28** | **79.74±0.37** |

Fed-RoD adopts an architecture similar to ours, utilizing both global and personalized classifiers to capture generalized and personalized information. However, it attempts to utilize representations at the same stage, posing challenges in simultaneously meeting these two contradictory objectives. In contrast, our proposed method strategically separates these two conflicting objectives into different stages of the model. This division allows us to achieve both generalization and personalization more effectively, ultimately resulting in superior performance across a wider range of scenarios.

## 5.5 Additional Analysis

**Comparison of global and personalized classifiers.** To gain a deeper understanding of the behavior of the global and personalized classifiers, we compare their accuracy, individually and in combination, during training. Figure 3 shows the corresponding experimental results on DomainNet. It is evident that personalized classifier significantly surpasses the global one, owing to its better alignment with local data distributions. Nevertheless, the accuracy of the local classifier can be significantly improved by combining its predictions with those from the global classifier. This enhancement is particularly notable in complex domains, such as Infograph. Conversely, in simpler domains like Quickdraw and Sketch, the benefit of combining classifiers becomes less pronounced. This occurs because, in simpler domains, the representations extracted by the personalized projection network are sufficient for each client's local tasks, thereby reducing the necessity for more diverse representations from the global encoder.

**Visualization of generalized and personalized representations.** To intuitively understand the generalized and personalized representations, we utilize t-SNE [39] for visualization. Figure 4 illustrates the visualization of both the generalized and personalized representations on DomainNet dataset. In these visualizations, different colors indicate different classes. It is noticeable that the personalized representations are more discriminative than the generalized ones, yet they exhibit lower consistency across clients. This demonstrates that DualFed can effectively separate the representation extraction process into two distinct stages, each characterized by high levels of generalization and personalization, respectively.

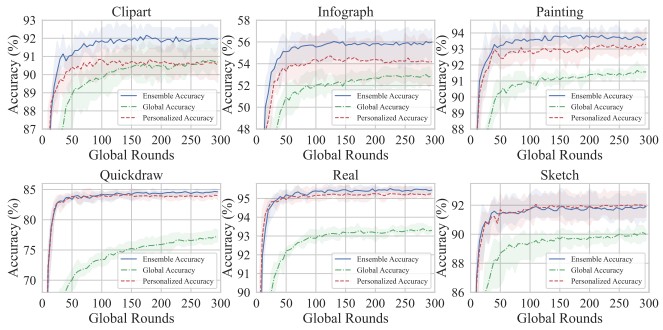

Figure 3: Test accuracy during training on DomainNet.

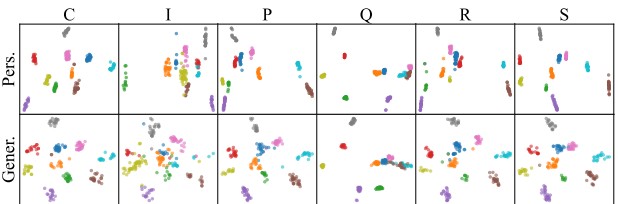

Figure 4: Visualization of representations on DomainNet.

**Quantitative evaluation of generalized and personalized representations.** We employ two metrics to quantitatively evaluate the evolution of generalized and personalized representations during training. To quantify the personaliztion of representations on clients, we adopt the class-wise *separation* in [25]. Additionally, we adopt the *linear centered kernel alignment (CKA)* [26], to measure the generalization ability of representation. Figure 5 presents the varying of *separation* during the training. The personalized representations can achieve higher class separation compared with the generalized representations. However, as shown in Figure 6, the similarity between clients of generalized representations is significant higher that that of the personalized representations.

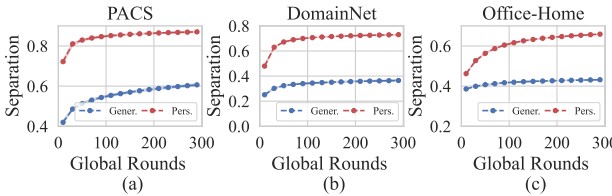

Figure 5: Class-wise separation during training.

**Comparison of Training Strategy.** DualFed employs a stage-wise training strategy, ensuring that the pre-projection representation remain undisturbed by specific local tasks, thereby maintaining its generalization. Here, we compare this training strategy with the one that training all parameters simultaneously. As shown in Table 4, when $E$ is relatively small (i.e., $E = 1$), simultaneous training can, in fact, outperforms stage-wise training. However, as $E$ increases (i.e., $E = 20$), simultaneous training lead to a obvious performance drop in PACS and DomainNet. This trend can be attributed to the fact that an increased number of local epochs causes

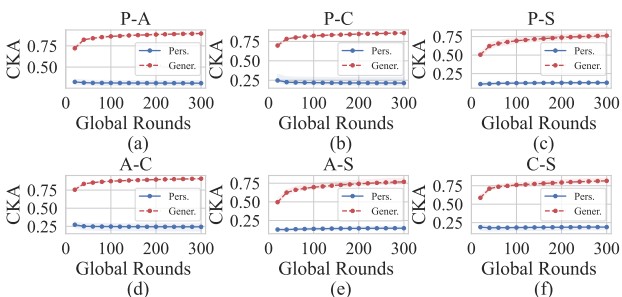

Figure 6: Client-wise CKA similarity during training.

the pre-projection representations to align more closely with the local task, thereby reducing their generalization.

Table 4: Experiments with Different Training Strategy.

| $E$ | Strategy | PACS | DomainNet | Office-Home |
|---|---|---|---|---|
| 1 | Stage-wise | 95.01±0.31 | 86.14±0.12 | 79.74±0.37 |
| | Simu. | 95.15±0.16 | 86.68±0.20 | 80.57±0.09 |
| 20 | Stage-wise | 94.17±0.28 | 84.49±0.18 | 75.93±0.77 |
| | Simu. | 93.85±0.30 | 84.71±0.33 | 75.42±0.65 |

**Effect of Projector Architecture.** We investigate the impact of the architecture of the projection network in three key aspects: the model depth ($D$), the dimension of hidden layers ($H$), the impact of BN layers. The corresponding results are shown in Table 5. While increasing $D$ can lead to more generalized pre-projection representations, it simultaneously reduces their discriminative power. Therefore, it is advisable to select an optimal $D$ that maintains a balance in the discriminative and generalized ability of the pre-projection representations. Increasing $H$ can enhance the model performance in most times. The importance of BN layers becomes more pronounced as the scale of the dataset increases.

Table 5: Experiments with Different Projector Architecture.

| $D$ | $H$ | BN | PACS | DomainNet | Office-Home |
|---|---|---|---|---|---|
| 1 | 256 | ✓ | 94.72±0.18 | 86.16±0.09 | 79.96±0.24 |
| 2 | 256 | ✓ | 95.01±0.31 | 86.14±0.12 | 79.74±0.37 |
| 3 | 256 | ✓ | 94.97±0.18 | 85.91±0.26 | 79.31±0.36 |
| 2 | 64 | ✓ | 95.35±0.19 | 86.06±0.32 | 79.43±0.24 |
| 2 | 128 | ✓ | 95.15±0.18 | 85.95±0.18 | 79.49±0.21 |
| 2 | 512 | ✓ | 95.21±0.17 | 86.23±0.23 | 79.97±0.35 |
| 2 | 256 | ✗ | 95.13±0.19 | 86.23±0.26 | 79.22±0.38 |

**Effect of Position of Global Classifier.** In DualFed, we employ a global classifier for generalized representations and a personalized classifier for personalized representations. Here we conduct experiments when placing the global classifier after the projector. In these experiments, we maintained a shared encoder and investigated two configurations: sharing the projection network (DualFed-G) and personalizing it (DualFed-P). As indicated in Table 6, removing the global classifier to the same stage as the personalized classifier

results in a significant performance decrease. This observation underscores the importance of the representations at different stages, as they provide complementary information that can enhance the overall performance of the model.

**Table 6: Experimental Results when Placing Global Classifier at Different Positions.**

|  | PACS | DomainNet | Office-Home |
|---|---|---|---|
| DualFed | 95.01±0.31 | 86.14±0.12 | 79.74±0.37 |
| DualFed-P | 94.95±0.18 | 85.55±0.09 | 78.24±0.29 |
| DualFed-G | 94.84±0.12 | 84.90±0.42 | 78.08±0.17 |

**Effect of Personalized Layers.** Table 7 presents the model performance with different personalization strategy. The results indicate that combining a global encoder with a personalized projection network significantly enhances model performance, as it integrates both generalized and personalized information.

**Table 7: Experimental Results with Different Parameter Personalized Strategies, where ✓ Denotes the Personalized Parameters, ✗ Denotes the Global Parameters.**

| Enc. | Prj. | P.C. | G.C. | PACS | DomainNet | Office-Home |
|---|---|---|---|---|---|---|
| ✗ | ✓ | ✓ | ✓ | 94.96±0.26 | 86.16±0.27 | 79.33±0.41 |
| ✗ | ✓ | ✓ | ✗ | 95.01±0.31 | 86.11±0.19 | 79.74±0.28 |
| ✗ | ✗ | ✗ | ✗ | 94.58±0.22 | 84.55±0.30 | 78.58±0.46 |
| ✗ | ✗ | ✓ | ✗ | 94.80±0.20 | 85.21±0.16 | 79.19±0.19 |
| ✓ | ✓ | ✓ | ✓ | 93.73±0.08 | 83.50±0.43 | 77.85±0.44 |

**Communication Costs.** We assess the communication costs by using the total number of model parameters transferred to reach a predefined target accuracy during training. For PACS, DomainNet, and Office-Home, the target accuracies are set to 85%, 75%, and 70%, respectively. As illustrated in Table 8, DualFed outperforms other methods by achieving the same target accuracy with lower communication costs, showcasing its practical efficiency.

**Table 8: Averaged Communication Costs (MB) when Reaching the Same Target Accuracy during Training.**

|  | PACS | DomainNet | Office-Home |
|---|---|---|---|
| FedAvg | 1920.93 | 2008.52 | 2538.72 |
| FedProx | 1833.62 | 2008.52 | 2538.72 |
| FedPer | 1658.47 | 1658.47 | 2007.62 |
| FedRep | 2705.92 | 3316.94 | 3753.37 |
| FedBN | 1482.91 | 1832.08 | 2098.97 |
| SphereFed | 3142.36 | 5586.42 | 18330.43 |
| Fed-RoD | 1135.10 | 1309.90 | 1663.30 |
| FedETF | 11783.85 | 15537.22 | 15973.66 |
| DualFed | **611.21** | **873.27** | **1225.59** |

**Effect of Hyper Parameters.** We conduct experiments using various hyperparameters, including the temperature coefficient ($\tau$),

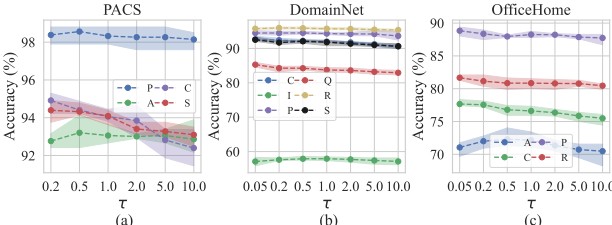

**Figure 7: Test accuracy with varying temperature coefficient.**

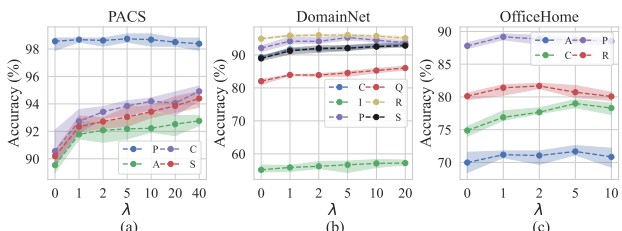

**Figure 8: Test accuracy with varying loss balance coefficient.**

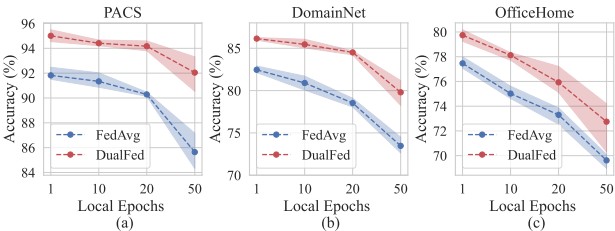

**Figure 9: Test accuracy with varying local epochs.**

the loss balance coefficient ($\lambda$), and the number of local epochs ($E$). As depicted in Figure 7, we observe that as $\tau$ increases, its effectiveness in distinguishing between different classes diminishes, thereby losing the advantage of contrastive loss. Figure 8 presents the test accuracy with varying $\lambda$. Setting $\lambda$ to 0 is equivalent to training the model solely with cross-entropy loss. Increasing $\lambda$ enhances the distinctiveness and relevance of personalized representations to the local task, which, in turn, improves model performance. However, as $\lambda$ value exceeding a certain threshold might cause training failures. Figure 9 shows the test accuracy with different local epochs, it illustrates that DualFed consistently surpasses FedAvg across various local epochs, demonstrating its robustness to local epochs.

## 6 CONCLUSION

In this paper, we have developed a new PFL approach called DualFed. DualFed decouples the objectives of generalization and personalization in PFL by a personalized projection network. This modification reduce the mutual interference between the conflicting optimization objectives in traditional PFL, thereby can achieve a win-win situation of both generalization and personalization in Non-IID FL. Our experiments across various datasets have shown that DualFed performs better than other FL methods, proving its effectiveness in handling the unique demands of PFL.

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
