# OpenReview forum: "DualFed: Enjoying both Generalization and Personalization in Federated Learning via Hierachical Representations"
_acmmm.org/ACMMM/2024/Conference — MM2024 Poster_

### Official Review · Reviewer_D7an · 2024-05-14

**Rating:** 4
**Confidence:** 3

**Summary:**

This paper introduces DualFed, aimed at enhancing both personalization and generalization in personalized federated learning. DualFed comprises pre-projection and post-projection representation stages. In the stage-wise training procedure for local clients, both the main branch and the global classifier undergo continuous training with global part aggregation, resulting in superior performance compared to other federated learning methods.

**Strengths:**

The paper devises a dual training process to distinctly address the generalization and personalization aspects of the model, effectively reconciling the conflicting objectives inherent in the task. Furthermore, the supplementary analysis section delves deeper into the architectural design and explores the incurred communication costs, providing a comprehensive understanding of the model's capabilities.

**Limitations:**

In this paper, certain design choices may require reevaluation or additional experimentation.

For instance, in the training strategy, the decision to keep the encoder trainable during pre- projection representation but frozen during post-projection representation may lack justification or require empirical validation. Additionally, exploring alternative ensembling methods for both the global classifier and personalized classifier, rather than directly summing them up, might need further investigation.

**Suitability:**

2

---

### Official Review · Reviewer_AMti · 2024-05-19

**Rating:** 3
**Confidence:** 3

**Summary:**

This paper presents DualFed, a novel approach in personalized federated learning (PFL) that addresses the challenge of balancing model generalization and personalization. By leveraging the hierarchical nature of deep models, DualFed introduces a personalized projection network between the encoder and classifier, effectively decoupling generalized and personalized representations. This design ensures minimal interference between these conflicting objectives, leading to improved performance. Extensive experiments on datasets such as PACS, DomainNet, and Office-Home demonstrate that DualFed significantly outperforms state-of-the-art FL methods, proving its effectiveness in handling Non-IID data distributions while maintaining efficiency in communication costs.

**Strengths:**

The strengths of the paper lie in its innovative approach to overcoming the conflict between generalization and personalization in PFL. By introducing a personalized projection network, DualFed effectively decouples these two objectives, leveraging hierarchical representations to optimize performance. This approach not only enhances the model's ability to generalize across clients but also allows for better adaptation to local data distributions. The extensive experimental results on multiple datasets, such as PACS, DomainNet, and Office-Home, showcase DualFed's superior performance compared to other state-of-the-art methods. Additionally, DualFed demonstrates robustness in handling varying local epochs and achieves these improvements with lower communication costs, making it a highly efficient and effective solution for federated learning in Non-IID scenarios.

**Limitations:**

1. Why not directly list the contributions in the introduction?

2. The paper organization is a little bit confusing. Why do you write the motivation in the preliminaries? When I read the introduction, I thought you have finished the motivation writing.

3. In figure 2, the forzen paratmers notations is different from the one in your picture.

4. Can you note the pre-projection representations and post-projection representations in Fig. 2?

5. According to the experiment, what is the relationship between your method and multimedia? Why not add some multimedia dataset in your experiment part?

6. It is hard to follow Table 5 because there are too many variables like D, H and BN.

7. In Fig. 8a,  AS C, A,S still perform better as the $\lambda$ get larger. This also happens in DomainNet. It is different from what you say as $\lambda$ value exceeding a certain threshold might cause training failures

7. Why, in Fig. 9, the local epochs get larger, the accuary performs worse?

**Suitability:**

1

---

### Official Review · Reviewer_Sfk2 · 2024-05-20

**Rating:** 5
**Confidence:** 2

**Summary:**

This paper introduces novel federated learning method that improves generalization and personalization at the same time via hierarchical representations. Extensive experiments demonstrate the proposed method is effective.

**Strengths:**

1.	Improving or balancing generalization and personalization is an important research problem in deep learning, it is novel to develop a method that tackles this problem in the challenging Non-IID FL setting.
2.	Leveraging the hierarchical representation to achieve both high model generalization and personalization at the same time is an interesting direction.

**Limitations:**

In general, the paper is well written, but I still have some questions regarding to this paper.
1.	There is one personalized classifier for each client, and one global classifier. What is the purpose of the global classifier? In what cases the output of the global classifier is different from the personalized classifier?
2.	In Eq. (7), the authors choose to use supervised contrastive loss. What if it used conventional supervised loss instead of supervised contrastive loss? Did you perform ablation studies to evaluate the difference?

**Suitability:**

2

---

### Official Review · Reviewer_JHPi · 2024-05-24

**Rating:** 2
**Confidence:** 3

**Summary:**

This paper proposes a new federated learning method DualFed. It aims to address the computational overhead issues of previous techniques that require choosing between universal or personalized representations. DualFed generates two types of representations: a general one before projection and a more personalized one after passing through a localized projection network in the encoder-classifier architecture. By using a shared encoder and training global and personalized classifiers concurrently, while combining their outputs at inference, DualFed can leverage both cross-client collaboration and personalization. Some experimental results are included in the paper to demonstrate the performance of the proposed method.

**Strengths:**

1. The method is simple and effective, able to meet the personalized needs of local clients and the generalized requirements of a central server.
2. Evaluations on mainstream datasets effectively demonstrate the effectiveness of the proposed method.
3. The overall organization of the paper is coherent and lucid, facilitating ease of comprehension.

**Limitations:**

1. It is crucial to clarify the differentiation between this paper and DBE[1]. Both of them use only linear layers to improve personalize task performance and DBE does not require two-stage training. It raises questions regarding the novelty of this work.
2. Lines 139-143 claimed that the method needs to maintain a shared encoder. However, some key information was not discussed. Such as which layers are personalized or generalized oriented, and how to determine the location where the projection layer is inserted?  The authors are encouraged to provide further explanation to enhance clarity.
3. The authors are advised to provide further explanation about the non-IID setting in experiments.

[1] Zhang J, Hua Y, Cao J, et al. Eliminating domain bias for federated learning in representation space[J]. Advances in Neural Information Processing Systems, 2024, 36.

**Suitability:**

2

---

### Meta-Review · Area_Chair_jp7J · 2024-07-09

**Recommendation:** Accept (Poster)
**Confidence:** 5

**Metareview:**

The paper presents DualFed, a novel federated learning approach aimed at balancing model generalization and personalization by introducing a personalized projection network in the encoder-classifier architecture. Reviewers acknowledged the innovative method and its potential to address the computational overhead in federated learning, as well as the clear and coherent presentation. However, concerns were raised regarding the novelty of the method compared to existing approaches, the clarity of certain methodological details, and the non-IID experimental settings. While some reviewers leaned towards acceptance due to the method's potential and thorough experimental validation, others highlighted significant unresolved issues, leading to a mixed consensus。

After carefully reading the paper and the comments, AC recommends acceptance since the organizational issue and inherent FL problems pointed out by some reviewers are not flaws of the paper, which slightly influence the rating. AC encourages the authors to read all comments and further improve the work.